# Validating a Non-Invasive Method for Assessing Cortisol Concentrations in Scraped Epidermal Skin from Common Bottlenose Dolphins and Belugas

**DOI:** 10.3390/ani14091377

**Published:** 2024-05-03

**Authors:** Clara Agustí, Xavier Manteca, Daniel García-Párraga, Oriol Tallo-Parra

**Affiliations:** 1Animal Welfare Education Centre (AWEC), School of Veterinary Science, Universitat Autònoma de Barcelona, 08193 Bellaterra, Spain; xavier.manteca@uab.cat (X.M.); oriol.tallo@uab.cat (O.T.-P.); 2Fundación Oceanogràfic de la Comunitat Valenciana, Research Department, Ciudad de las Artes y las Ciencias, 46013 Valencia, Spain; dgarcia@oceanografic.org

**Keywords:** cortisol, skin, stratum corneum, keratin, steroid hormones, cetacean, stress, conservation biology

## Abstract

**Simple Summary:**

Our society is increasingly concerned about the well-being of animals like dolphins, which can be affected by life under professional human care or by anthropogenic disturbances in the wild. To better understand and improve their welfare, scientists are exploring new, non-invasive methods to study the response to stressors in these animals. This research introduces an approach for measuring the stress response by analysing cortisol from epidermis samples, a method that does not require invasive procedures. We developed and tested a reliable technique to extract and measure cortisol levels from the epidermis of common bottlenose dolphins and belugas. Our findings show that this method works well and is accurate, even with very small epidermis samples. We also found that the amount of stress hormones can vary depending on the individual animal but not necessarily where the body sample was taken. This study is a step forward in non-invasively studying and enhancing the welfare of cetaceans, offering insights into their stress levels in a way that is safer and more comfortable for them, which could ultimately lead to better care and conservation practices for these species.

**Abstract:**

Society is showing a growing concern about the welfare of cetaceans in captivity as well as cetaceans in the wild threatened by anthropogenic disturbances. The study of the physiological stress response is increasingly being used to address cetacean conservation and welfare issues. Within it, a newly described technique of extracting cortisol from epidermal desquamation may serve as a non-invasive, more integrated measure of a cetacean’s stress response and welfare. However, confounding factors are common when measuring glucocorticoid hormones. In this study, we validated a steroid hormone extraction protocol and the use of a commercial enzyme immunoassay (EIA) test to measure cortisol concentrations in common bottlenose dolphin (*Tursiops truncatus*) and beluga (*Delphinapterus leucas*) epidermal samples. Moreover, we examined the effect of sample mass and body location on cortisol concentrations. Validation tests (i.e., assay specificity, accuracy, precision, and sensitivity) suggested that the method was suitable for the quantification of cortisol concentrations. Cortisol was extracted from small samples (0.01 g), but the amount of cortisol detected and the variability between duplicate extractions increased as the sample mass decreased. In common bottlenose dolphins, epidermal skin cortisol concentrations did not vary significantly across body locations while there was a significant effect of the individual. Overall, we present a contribution towards advancing and standardizing epidermis hormone assessments in cetaceans.

## 1. Introduction

Societies show a growing concern about the welfare of cetaceans maintained in captive settings worldwide. Further, this concern spreads to free-ranging animals threatened by anthropogenic disturbances and environmental changes and whose habitats are under human management actions. In this context, adapting and developing tools for a scientific animal welfare assessment has become an emerging necessity [1,2,3].

Animal welfare is a multidimensional concept that must be assessed through different approaches and methodologies. Among these, the physiological parameters related to health and emotional states, and more specifically to the physiological stress response, are useful “nonspecific” indicators of a variety of potential welfare problems [4,5]. Exposure to stressful stimuli usually results in an increased secretion of glucocorticoid hormones (GCs) subsequent to the activation of the hypothalamic–pituitary–adrenal (HPA) axis [6]. Importantly, the stress response is not inherently bad as it helps an animal to cope with its environment and challenging circumstances. However, if activated excessively or for a long time it may have adverse effects resulting in impaired biological functions (e.g., reproduction, immunity, and growth [7]).

Cortisol is one of the main GCs secreted in marine mammals, and it has been identified as a potential biomarker of the stress response in several cetacean [8,9,10]. Cortisol in both captive and wild cetaceans has been measured in multiple matrices such as serum or plasma [11,12,13], saliva [14], respiratory vapour (‘blow’) [15], urine [11], faeces [12], and blubber [12,15,16,17].

Although blood is the typical matrix used to measure cortisol, alternative tissues offer advantages as most are less invasive and enable hormonal assessments in free-ranging cetaceans without the need to capture them. Moreover, alternative matrices in endocrine assessments allow different assessments of the stress response of a single individual. For instance, marine mammal’s blubber and faeces have been shown to reflect relatively recent elevations in serum cortisol, being detectable within two and five hours, respectively, in common bottlenose dolphins [12], and thus, they have been proposed as indicators of the mid-term activation of the HPA axis. In contrast, cortisol levels accumulated in keratinous materials have been considered an indicator of the longer-term (‘chronic’) activation of the HPA axis [18]. Cortisol concentrations have been measured in the hair and nails of non-cetacean mammals (e.g., [19,20,21,22]), feathers of birds [23,24], and shed skins of snakes [25]. These keratinous matrices are not present in cetaceans, but it has been possible to measure long-term accumulated cortisol in the epidermis [26], baleen [27], and earplug [28].

Recent research by Bechshoft et al. [26,29] indicates that the epidermis could be adequate to assess long-term stress in living cetaceans. This would be particularly helpful in conservation research for assessing chronic cumulative impacts and establishing relationships between the cause and effect, among others [30,31].

Skin is the body’s physical barrier between the external and the internal environments and communicates with neurological, endocrine, and immune regulatory networks. As in other mammals, cetacean skin is composed of epidermal, dermal, and hypodermal layers [32]. However, its anatomy and functions have been highly modified from its terrestrial ancestors [33].

Cetacean epidermis is thicker than that of terrestrial mammals, allowing higher resistance and the maintenance of homeostasis in water [34]. It is parakeratotic and lacks the stratum granulosum and stratum lucidum [35]. It consists of three histologically distinct layers: the stratum basal (that generates new cells), the stratum intermedium, and the stratum corneum [36]. The basal layer has continuous mitoses and sloughing, and desquamation caused by water friction is rapid, showing a very high cellular turnover rate [37]. In common bottlenose dolphins the epidermal turnover time has been described as lasting approximately 73 days [37].

The growth and replacement of cetacean epidermis is understood as a continuous process [33] affected by factors such as trauma, hormonal influences, diurnal effects, and environmental fluctuations in temperature [37]. Belugas, however, have a unique pattern of habitat use that has set the stage for cyclical ‘phases’ in the epidermal growth, which have been described analogous to a molt [38]. Studies have documented increased thyroid secretion [39] and enhanced epidermal cell proliferation [38] at the time of year when belugas occupy estuaries. There, the exposure to warm fresh water and active abrasion of the skin surface may promote the proliferation of new skin cells as well as the detachment of cellular debris from the stratum externum [40].

Cortisol concentrations have been successfully measured in epidermal skin samples from harbour porpoise (*Phocoena phocoena*; Bechshoft et al. [26]) and common bottlenose dolphins [29] through liquid chromatography–mass spectrometry (LC—MS/MS). Peaks in circulating cortisol (i.e., acute stress) have been detected in common bottlenose dolphin sloughed epidermis with an average delay of 46 days, suggesting that full-depth epidermis samples could reflect longer-term cortisol levels (i.e., chronic stress) during that period of time [29]. Nevertheless, before using epidermal cortisol concentrations as an indicator of stress and therefore as a tool to assess cetacean welfare, several knowledge gaps need to be addressed. Some of these are related to methodological considerations while others are related to the kinetics of cortisol integration into skin [27,41].

Confounding factors are common when measuring glucocorticoid hormones in different matrices [19,42,43,44,45,46], and identifying and characterizing these is necessary to interpret these indicators correctly. In particular, the interpretation of cetacean epidermal cortisol concentrations may be confounded by sample storage and treatment techniques, the sex and age of the individual, variations in matrix growth rates, the location of the body and the epidermal layers sampled, among others [44,46,47,48,49]. Further, obtaining cetacean skin samples can be challenging, and especially in free-ranging animals, samples are often divided for multiple analyses in order to maximize their usefulness. For this reason, establishing the minimum sample mass needed to obtain robust, replicable measurements of cortisol concentrations for an extraction method is essential [50].

Using an immunoassay is probably the most common method for analysing GC levels in diverse tissue types, and two versions are mostly used, enzyme immunoassays (EIAs) and radioimmunoassays (RIAs), both competitive binding assays and highly sensitive. Importantly, each assay must be validated for a new species and matrix to ensure reliable and interpretable results [51]. Furthermore, it is noteworthy that liquid chromatography–tandem mass spectrometry (LC-MS/MS) has been widely adopted, especially in recent research, for quantifying hormones in non-invasive samples from across diverse species.

Our objectives in this study were (i) to develop a method for a non-invasive epidermis desquamation collection in captive common bottlenose dolphins and belugas; (ii) to validate a protocol for steroid hormone extraction and for the analysis of cetacean epidermal cortisol concentrations using a commercial EIA test; and (iii) to identify potential confounding factors that could affect cortisol concentrations related to sex, sample mass, and body location.

## 2. Materials and Methods

### 2.1. Experimental Design

Ten common bottlenose dolphins and three belugas housed in the Oceanogràfic de Valencia aquarium at the City of the Arts and the Sciences (Comunidad Valenciana, Spain) were used in this study. Common bottlenose dolphins included in the study were eight adults (four males and four females) and two juveniles (males) with a mean age of 23.8 years (range 9 to >40). Belugas included one calve male, one adult female, and one adult male with an age of 3, 22, and more than 55 years, respectively.

The dolphin’s facility consisted of seven outdoor pools eleven metres deep and interconnected by variably opened or closed gates, with a total capacity of 23 million litres of sea water. Water temperatures ranged from 19.21 to 26.26 °C during the year, ensuring that all animals were always within their thermal comfort range. Diets consisted of frozen fish, mainly herring (*Clupea harengus*), capelin (*Mallotus villosus*), hake (*Merluccius merluccius*), and squid (*Loligo spp.*), and were formulated to meet individual animal requirements. The beluga’s facility consisted of four indoor pools six metres deep and interconnected by variably opened or closed gates, with a total capacity of 3.5 million litres of sea water. Water temperature varied between 11 and 16 °C. Diets consisted of frozen fish, mainly herring (*Clupea harengus*), capelin (*Mallotus villosus*), hake (*Merluccius merluccius*), sprat (*Sprattus sprattus*), and blue whiting (*Micromesistius poutassou*), and were formulated to meet individual animal requirements.

Positive reinforcement training was the main tool used to assist with animal husbandry, veterinary, and research procedures. Epidermis sampling was conducted without causing injury to the animals and without altering their daily programmed activities, locations, or group compositions. Moreover, individuals attended and participated voluntarily in the sampling; otherwise, the procedure was postponed.

### 2.2. Epidermal Sampling

One sample per individual was collected weekly over the course of a year between 2018 and 2019. Epidermis sampling consisted of a trainer placing the animal in a “line up” position (parallel to the edge of the facility) and with the sampling area out of the water. New behavioural training was not necessary as the animals were desensitized with different objects and techniques as a part of their daily routine. The epidermis sampling area was prevented from getting wet and dried with a gauze pad. A semi-rigid plastic card, comparable in material and flexibility to an identity document or a membership card, featuring a smooth edge and sterilized with alcohol, was utilized to gather desquamated epidermis. This was achieved by scraping in multiple directions and applying moderate pressure on healthy skin areas approximately 15 × 15 cm in size (Figure 1). After 3 to 6 scrapings, the sampled epidermis was transferred into a 1.5 mL Eppendorf tube. This was achieved by carefully aligning the edge of the card with the tube’s aperture and guiding the epidermis particulates into the tube, all while wearing gloves to prevent any contamination (Figure 1). After 3 to 6 scrapings, sampled epidermis flakes were transferred into a 1.5 mL Eppendorf tube by meticulously guiding the edge of the card into the tube’s aperture without direct contact and employing gloves to prevent contamination. Subsequently, the sample was transported to the laboratory, accompanied by refrigerant gel packs within a range of 5 to 15 min, and was then stored at −20 °C to ensure preservation.

### 2.3. Sample Preparation and Storage

Frozen skin samples were dried in an oven (Heraeus model T6; Kendro^®^ Laboratory Products, Langenselbold, Germany) at 36 °C for 72 h in order to evaporate the remaining water. Once dried, samples were cut into small pieces and then ground in a ball mill (MM200, Retsch, Haan, Germany) for 15 min at 25 Hz to homogenize the contents. To avoid losing samples, each Eppendorf tube with stainless-steel balls inside was fixed inside a 10 mL stainless-steel grinding jar. Grinding media was then separated using magnets, and the pulverized samples were stored at −20 °C.

### 2.4. Epidermis Pool, Effect of Sample Mass, and Sample Processing Error

We created homogenous mixtures of pooled epidermis samples to perform validation tests and examine methanol extraction efficiency with different amounts of samples. For common bottlenose dolphins, five pools were created using between 4 and 12 samples that were previously homogenized and pulverized. This epidermis dust was then mixed thoroughly. Pooled samples were then divided into 10 subsamples in five duplicated mass categories: 5, 10, 20, 50, and 100 mg (solvent/sample ratios of 300:1, 150:1, 75:1, 30:1, and 15:1 (μL:mg), respectively). For belugas, three pools (one per individual) were created using between 3 and 7 samples as described before. Pooled samples were then divided into 6 subsamples in three duplicated mass categories: 10, 50, and 100 mg (solvent/sample ratios of 150:1, 30:1, and 15:1 (μL:mg), respectively). Each subsample was extracted and processed in the laboratory independently.

### 2.5. Effect of Body Location

To evaluate the effect of body location, all individuals were sampled simultaneously on multiple body sites. The objective was to examine whether significant differences in epidermal cortisol concentrations existed depending on the region of the body from which the sample is collected.

Dolphins’ scraped epidermis samples were collected from eight locations along the left and right side of the individuals: (1) left dorsal fin; (2) right dorsal fin; (3) left dorsal peduncle; (4) right dorsal peduncle; (5) left ventral peduncle; (6) right ventral peduncle; (7) dorsal caudal fin; and (8) ventral caudal fin (Figure 2). Samples were normalized by a mass of 15 to 20 mg due to an effect of sample masses detected in the previous phase of the study. Samples of less than 15 mg were discarded from analysis. Finally, samples were dried and stored, as described in 2.3, before hormone extraction.

Belugas’ scraped epidermis samples were collected from ten locations along the left and right side of the individuals: (1) left dorsal anterior to the dorsal ridge; (2) right dorsal anterior to the dorsal ridge; (3) left dorsal ridge; (4) right dorsal ridge; (5) left dorsal peduncle; (6) right dorsal peduncle; (7) ventral peduncle; (8) dorsal caudal fin; (9) ventral right caudal fin; and (10) right dorsal pectoral fin. In all three individuals, we only obtained enough samples (equal or more than 15 mg of dry epidermis) in the dorsal caudal fin and in the dorsal pectoral fin. Due to the low number of individuals and samples obtained, we decided to use the right dorsal pectoral fin as the standard sample location throughout the study without analysing the differences between the two locations.

### 2.6. Hormone Extraction

A methanol-based extraction protocol was designed based on the method previously described by Tallo-Parra et al. [52] to extract cortisol from the hair of dairy cows. For each sample, 1.5 mL of pure methanol was added, and the samples were vortexed and then moderately shaken for 18 h at 30 °C (G24 Environmental Incubator Shaker; New Brunswick Scientific Co. Inc., Edison, NJ, USA) for steroid extraction. Following the extraction, samples were centrifuged at 7000× *g* for 2 min.

Subsequently, 1.2 mL of supernatant was transferred into a new 2 mL Eppendorf tube and then placed in an oven (Heraeus model T6; Kendro Laboratory Products, Langenselbold, Germany) at 38 ºC. Once the methanol was completely evaporated (approximately after 36 h), the dried extracts were reconstituted with 0.15 mL of EIA buffer (1 M phosphate buffered saline) provided by the EIA assay kit (Cortisol ELISA Kit; Neogen^®^ Corporation, Ayr, UK) and vortexed for 30 s. This dilution was chosen to fall near the 50% bound on the standard curve, the area of greatest assay precision. Then, the samples were immediately stored at −20 °C until analysis.

### 2.7. Hormone Detection and Assay Validation

Cortisol concentrations and validation tests were determined by using three competitive EIA kits (Neogen^®^ Corporation Europe, Ayr, UK) and following the manufacturer’s instructions. Standard curves ranged between 0 and 3.8 ng/mL, and cortisol concentrations in the samples were determined using a linear regression model based on the standard curve. All samples were assayed in duplicate, and the mean hormone concentration was recorded.

Following the essential criteria for immunological validation [53], the precision, specificity, accuracy, and sensitivity of the assays were determined. Extracts from pooled 20 mg samples for common bottlenose dolphins and 50 mg samples for belugas were used for both the assay validation and the study of sample mass effect.

Precision was evaluated by calculating intra- and inter-assay coefficients of variation (CV). The intra-assay CV was calculated as a mean of the intra-assay CV of all the samples analysed per duplicate. The inter-assay CV was calculated only for common bottlenose dolphins’ samples as a mean of the inter-assay CV of 2 pooled samples analysed per duplicate in two EIA kits.

Accuracy was assessed through the spike-and-recovery test by adding known volumes of pooled extracts to different known concentrations of pure standard cortisol solution. Then, recovery was calculated to examine the possible interference of components within the extract with antibody binding. The percent recovery was calculated using the following formula: (amount observed/amount expected) × 100. The amount observed was the value obtained though the EIA analysis, and the amount expected was the mathematical calculation of cortisol concentrations in the spiked sample considering the original concentrations of both pooled extracts and standard cortisol solutions.

Specificity was assessed by the linearity of the dilution, determined by using 1:1, 1:2, 1:5, and 1:8 dilutions of pools with EIA buffer. According to the manufacturer, the cross-reactivity of the EIA antibody with other steroids is as follows: prednisolone 47.4%, cortisone 15.7%, 11-deoxycortisol 15.0%, prednisone 7.83%, corticosterone 4.81%, 6β-hydroxycortisol 1.37%, 17-hydroxyprogesterone 1.36%, and deoxycorticosterone 0.94%. Steroids with cross-reactivity less than 0.06% are not presented. Finally, sensitivity was given by the smallest amount of hormone concentration detected.

### 2.8. Statistical Analyses

Data were processed and analysed using Statistical Analysis System (SAS.9.4. software, SAS Institute Inc.; Cary, NC, USA). All the values are presented as mean ± SD. A *p*-value < 0.05 was considered for significance.

For the biochemical validation, statistical correlations in the dilution test (expected vs. obtained values) was determined using the Pearson’s Product correlation test. One-way analysis of variances (ANOVAs) were performed to determine if there were significant differences in the obtained sample mass between individuals, sampling week (including one year from January to December in the analysis) and season. Independent sample t-tests were used to determine significant differences between common bottlenose dolphins’ sexes and ages (adults and juveniles) in the amount of obtained sample mass. Linear regression was performed to test for relationships between the sample mass and epidermal cortisol concentrations.

Finally, a mixed linear regression model (PROC MIXED, the method of restricted maximum likelihood (REML)) was applied to investigate the effect on dolphins’ epidermal cortisol concentrations of body location, sex, age, and ELISA plate, included as fixed effects. Each individual was treated as a random effect in the model.

## 3. Results

### 3.1. Validation of the Epidermis Collection Methodology and the EIA

In common bottlenose dolphins, we collected 407 samples of 35.4 ± 23.18 mg of dry epidermis, range: 1 to 145 mg. The semi-rigid plastic card consistently collected enough sample mass (≥20 mg of dry epidermis) in 72.97% of the sampling attempts (Figure 3). Significant differences in the dried sample mass were found among individuals (ANOVA: F(9, 397) = 6.04, *p* < 0.001), sampling weeks (ANOVA: F(45, 361) = 3.49, *p* < 0.001), and seasons (ANOVA: F(3, 403) = 12.16, *p* < 0.001). Post hoc Tukey HSD tests revealed that fall (46.14 ± 27.28 mg) had a significantly higher sample mass compared to summer (28.70 ± 16.86 mg) and winter (32.94 ± 23.44 mg). There were no significant differences between fall and spring (34.36 ± 20.13 mg) or among spring, summer, and winter. No significant differences in the residuals of the sample mass were found between sexes (*t*-test: t(405) = 0.85, *p* = 0.398). However, the sample mass tended to be higher in adult (37.89± 23.56 mg) than in juvenile individuals (30.98 ± 21.87; t-test: t(405) = 2.92, *p* < 0.01).

In belugas, we collected 117 samples of 113.03 ± 213.28 mg of dry epidermis, range: 3 to 1076 mg. The semi-rigid plastic card consistently collected enough sample mass (≥ 20 mg of dry epidermis) in 94.87% of the sampling attempts (Figure 3). No significant differences in the residuals of the sample mass were found among individuals (ANOVA: F(2, 114) = 0.94, *p* = 0.393), sampling weeks (ANOVA: F(52, 64) = 0.947, *p* = 0.333), nor across seasons (ANOVA: F(3, 113) = 1.43, *p* = 0.239).

The individuals of both species participated voluntarily in all sampling attempts and did not show avoidance or discomfort behaviours. The sampling time per individual was around 1 min.

For common bottlenose dolphins, the mean intra-assay CV was 9.35 ± 7.13%. The mean inter-assay CV was 4.55 ± 3.65%. In the linearity of the dilution, the obtained cortisol concentrations were correlated with the expected cortisol values (Pearson: r(3) = 0.96, *p* = 0.017; Figure 4). The average recovery percentage from the spike recovery test was 108.17 ± 20.85%. The sensitivity of the assay was 0.061 ng cortisol/g of dried epidermis.

For belugas, the mean intra-assay CV was 8.06 ± 5.33%. In the linearity of the dilution, the obtained cortisol concentrations were correlated with the expected cortisol values (Pearson: r(3) = 0.98, *p* = 0.017; Figure 4). The average recovery percentage from the spike recovery test was 115.17 ± 15.86%. The sensitivity of the assay was 0.03 ng cortisol/g of dried epidermis.

### 3.2. Effect of Sample Mass

In common bottlenose dolphins, the measured concentrations of cortisol per gram of dry epidermis increased significantly as the masses of the pooled samples decreased (linear regression model: R2 = 0.63, F(1, 48) = 82.51, *p* < 0.001). Moreover, the one-way ANOVA indicated a significant effect of the sample mass on epidermal cortisol concentrations (F(4, 45) = 49.64, *p* < 0.001). The post hoc Tukey’s honestly significant difference (HSD) test revealed significantly higher cortisol concentrations (ng/g) in the 5 mg sample mass compared to those in the 10 mg (HSD: *p* = 0.004), 20 mg, 50 mg, and 100 mg sample masses (HSD: *p* < 0.001; Figure 5). The 10 mg sample mass exhibited lower cortisol levels than the 5 mg samples (HSD: *p* = 0.004) but higher levels than those in the 20 mg (HSD: *p* = 0.005), 50 mg, and 100 mg sample masses (HSD: *p* < 0.001; Figure 5). Additionally, cortisol extractions from 50 mg samples showed concentrations comparable to those from 20 mg and 100 mg samples (HSD: *p* = 0.97 and 0.121, respectively; Figure 5).

In belugas, the measured concentrations of cortisol per gram of dry epidermis increased significantly as the masses of the pooled samples decreased (linear regression model: R2 = 0.42, F(1, 16) = 11.49, *p* = 0.004). However, the one-way ANOVA did not find an effect of the sample mass on epidermal cortisol concentrations (F(2, 15) = 0.24, *p* = 0.783).

Moreover, variability between duplicate extractions was, in most cases, high for both 5 and 10 mg samples and low for 20, 50, and 100 mg samples (Figure 5).

### 3.3. Effect of Individual, Body Location, and Sex in Common Bottlenose Dolphins

The average epidermal cortisol concentration was 0.71 ± 1.05 ng cort/mg of dry epidermis (range: 0.13 to 8.09).

Epidermal cortisol concentrations did not vary significantly across body locations (LMM: fixed effect; F = 0.84; *p* = 0.568; Figure 6), sexes (LMM: fixed effect; F = 0.44; *p* = 0.529), nor EIA plate (LMM: fixed effect; F = 0.71; *p* = 0.428). Meanwhile, there was a significant effect of the individual (LMM: random effect; *p* < 0.05; Figure 7).

## 4. Discussion

In this study, we analysed cortisol concentrations in the epidermis of common bottlenose dolphins and belugas to validate a method for epidermis collection and steroid hormone extraction, as well as the use of an enzyme immunoassay (EIA) test to quantify cortisol concentrations in this alternative matrix.

### 4.1. Validation of the Epidermis Collection Methodology and the EIA

The use of a semi-rigid plastic card provided samples to measure cortisol in most sampling attempts without any apparent discomfort for the animals and without the need of complex training or altering the facility routines. Thus, it proved to be a non-invasive, easy, safe, and fast (i.e., feasible) method to collect samples from this species in captivity. However, amplifying the epidermis sampling area to more than ≈15 × 15 cm in common bottlenose dolphins may have provided a bigger proportion of samples complying with the minimum sample mass required to perform cortisol analysis in the present study.

In free-ranging animals, obtaining epidermis samples through these methods would be possible when animals are restrained for tagging or health assessments and during disentanglement efforts. Moreover, epidermis samples can be obtained by remote biopsying [54] or even by collecting the epidermis naturally sloughed off at sea surface [55] or from animals bow riding boats. The epidermis from stranded animals and tissue banks could also be used, enabling retrospective studies [26].

The amount and appearance of the epidermis collected varied among the samples from both species. For instance, some sampling attempts resulted in trace amounts of epidermis collected, while others resulted in a mixture of small particles of sloughed epidermis or in peeled sheets. Therefore, when designing studies, it is important to consider that sampling in captive settings may be sometimes unsuccessful. This is also described in Bechshoft et al.’s study [29] which suggests that sloughing is not continual but occurs in pulses with different stages of epidermis turnover. In common bottlenose dolphins, the amount of collected epidermis significantly varied among individuals, weeks, and seasons. Individual variability was related to age but not to sex. The variability in the amount of epidermis collected may correspond to differences in skin cell proliferation and maturation rate as a function of skin trauma and hormonal influences (individual and sex variation) and as a function of environmental fluctuations in temperature and salinity (week and season variation; Hicks et al., 1985). However, additional research characterizing skin growth is necessary to clarify these variations.

Contrarily, in belugas, the amount of collected epidermis did not vary among individuals, sampling weeks, nor seasons. However, we had a low sample size of three individuals. Moreover, although belugas show seasonal patterns of epidermal growth, a reduction in the circulating levels of thyroid hormones [56] and the absence of strong seasonal or environmental cues under human care may suppress epidermal growth cycles in captive individuals.

The assay validation results indicate that cetacean desquamated epidermis contains a quantifiable amount of cortisol that can be detected with the methodology presented here, even using an EIA kit not designed specifically for the epidermis nor cetacean species.

Both intra-assay CVs revealed good repeatability within the assays, while the inter-assay CV suggested a good repeatability between the two assays. In the spike-and-recovery test, various quantities of cortisol previously added to the pooled extracts were quantitatively recovered; thus, other components of the samples probably did not interfere acutely with the estimation of the hormone. Meanwhile, the serial dilution of pooled scraped epidermis sample extracts ran parallel to both assay standard curves, suggesting that cortisol successfully bound to the antibody in a dose-dependent way and without other substances in the epidermis matrix interfering with the steroid–antibody interaction. However, we cannot dismiss the possibility that uncommon steroid metabolites and/or conjugated steroids were present in dolphin or beluga epidermis and, not included in the manufacturer’s information, cross-reacted with the antibody [57].

The use of high-performance liquid chromatography coupled with tandem mass spectrometry (LC-MS/MS) as described in Bechshoft et al. [26,29] is a more precise and sensitive technique than EIA for the quantification of epidermal cortisol concentrations [57]. However, in contrast to LC-MS/MS, EIA equipment is significantly more economical and demands lower analytical skills [57]. For this reason, immunoassays remain the method of choice in many laboratories and in particular of researchers studying wildlife.

Finally, the range of cortisol concentrations obtained in common bottlenose dolphins (0.13 to 8.09 ng cort/mg of dry epidermis) is comparable to that obtained in the same species by Bechshoft et al. [29], who reported range values of 0.31 to 16.17 ng cort/mg of dry epidermis. This may suggest that the modifications made to the previously described methodology by Bechshoft et al. [26] have a minimum impact on hormone extractions and quantifications. Additionally, belugas exhibited a cortisol concentration range (0.47 to 1.44 ng cort/mg of dry epidermis) comparable to that of common bottlenose dolphins.

### 4.2. Effect of Sample Mass

The use of very small epidermis samples (e.g., 5 and 10 mg) resulted in higher apparent cortisol concentrations in both species. In common bottlenose dolphins, samples of 5 and 10 mg resulted in a higher dispersion of the values and variability between duplicate extractions and repeated measurements in the EIA. This effect has already been documented for cortisol in other matrices such as blubber [58], feathers [59], and faeces [44], and it could be caused by several methodological problems. For instance, errors associated with mass weighting have a stronger quantitative impact on the final calculation of hormones when the sample mass decreases [47,60]. Moreover, the efficiency of cortisol extraction may decline at lower extract-solvent-volume-to-sample-mass ratios [47].

Our results suggest a potential sample mass threshold of 20 mg, below which cortisol concentrations data seem to become overestimated and less repeatable. Therefore, we recommend that future studies use this protocol to avoid extractions of small samples (<20 mg). Moreover, given the variability in apparent cortisol concentrations between sample mass classes, we suggest standardizing the sample mass used throughout studies whenever possible. In our study, we used a standard sample mass of 20 mg in common bottlenose dolphins as mass values higher than 20 mg can be difficult to obtain (for instance, 72.97% vs. 23.59% of valid samples for 20 mg and 50 mg of sample mass, respectively). In belugas, we initially decided to use a standard sample mass of 50 mg for the methodological validation. However, only 53% of the collected samples reached this dry mass, while 94.87% of the samples reached 20 mg. Therefore, we recommend amplifying the sampling area or using a standard sample mass of 20 mg in both species.

### 4.3. Effect of Individual, Body Location, and Sex

The individual had a significant effect on epidermal cortisol concentrations. Although this study was intended to be the first step in the validation of epidermal cortisol concentrations as an indicator of the stress response in common bottlenose dolphins, the inter-individual differences observed could be related to differences in individuals’ endocrinological status. Meanwhile, no differences in epidermal cortisol concentrations were found among sexes. This coincides with other studies showing no differences in cortisol concentrations across sexes in common bottlenose dolphin blood [11,42], faeces [61], and blubber [62]. Conversely, other authors detected sex-related variations in dolphin blood [15], while many studies across vertebrate taxa have found a variation in cortisol levels with respect to sex (e.g., [63,64,65]). This may be partly explained by sex differences in the body condition index [66] or in the sex-specific effects of gonadal steroids on basal and stress-induced HPA axis activity [67,68]. Further studies with higher sample sizes are needed to assess the influence of sex on epidermal cortisol concentrations.

Although differences in epidermal cortisol concentrations were not found among dolphin body locations in this study, these results should be interpreted with caution, particularly due to the limited number of animals sampled. In fact, the inconsistent but high variability obtained in epidermal cortisol concentrations between body locations could suggest that cortisol is not homogeneously distributed along dolphin epidermis.

The ventral part of cetacean epidermis is typically thicker [69], while differences across body regions exists in the dermal papilla height [70] and colour due to regional differences in the concentration of melanocytes [71,72]. The heterogeneity in these and other potential traits may explain cortisol regional variations along the body epidermis, which may lead to erroneous results or comparisons between studies. For instance, in dogs and chimpanzees, darker hairs had lower cortisol concentrations than lighter ones [73,74,75], whereas the opposite was true in grizzly bears and dairy cattle [19,52]. The body location of blubber has been related to differences in steroid hormone concentrations in cetaceans [49,76,77]. Considering these observations, we suggest that sampling the same specific body locations would facilitate a more nuanced analysis in hormonal trends.

### 4.4. Cetacean Epidermis as a Storage Medium of Steroid Hormones

Epidermis growth occurs in the basal layer of the epidermis where matrix cells (keratinocytes and melanocytes) undergo proliferation cycles that provide for the renewal of the tissue. During epidermis growth, newly formed cells constantly displace the older cells upward, first to the stratum spinosum and subsequently to the epidermis surface, which consists of a stratum corneum with a parakeratosis-like morphology [35]. Finally, the stratum corneum is sloughed off to the environment as sloughed epidermis [36]. The speed of skin renewal may show some variation between individuals, depending on factors such as trauma, hormonal influences, diurnal effects, and environmental fluctuations in temperature and salinity. However, a period of approximately 73 days for the migration of cells from the basal lamina to the most external surface has been described [37].

Cortisol synthesis occurs in pulsatile events in the adrenal gland followed by infusion into the bloodstream [78]. The precise mechanisms by which cortisol is incorporated from blood into growing skin cells is still not understood. However, following the multicompartment model [79,80] and due to cortisol’s lipophilic character, the most likely incorporation route is passive diffusion. Moreover, according to the free hormone hypothesis [81], only the unbound, the free cortisol fraction of the plasma would be incorporated into skin.

Keratinous (or in case of skin, lipokeratinocytic; [82]) tissues are thought to reflect the long-term cortisol status since its molecules are incorporated into cells as they grow and keratinize (e.g., [83,84]. Evidence suggests that both the baleen and earplug from whales can trap hormones, leaving a historical record of physiological state fluctuations [27,28,85]. For this reason, as is the case for the abovementioned matrices, it has been postulated that a full-depth epidermis sample may reflect chronic cortisol levels, while a section of the stratum basal (where cells are constantly produced) would most likely reflect more recent circulating cortisol concentrations, and a section of the stratum externum and sloughed epidermis would provide information on the oldest physiological status [29]. In fact, the relationship found in common bottlenose dolphins between cortisol peaks in blood, blubber (2 h delay compared to blood), and epidermis (45–60 days delay compared to blood) provides evidence of the latest [29].

However, in contrast to the baleen or earplug, the epidermis is a metabolically active tissue, which demands caution in interpreting the measured hormone concentrations retrospectively in relation to its growth rate. For instance, although the measurement of glucocorticoids in hair has been presented as a method for assessing chronic stress in studies on humans and animals, some authors have questioned the validity of hair cortisol as a marker of stress outside the immediate past [86]. In some species, hair cortisol concentrations seem to increase rapidly in response to acute stress events [66,87]. Further, recent research using radio-labelled cortisol suggest that the hormone may be displaced along the hair shaft and converted to cortisone and other metabolites instead of forming pure cortisol discrete bands [88].

Therefore, as is the case with hair [86], we suggest there is no evidence at all to assume that cortisol molecules accumulate and remain permanently locked into the skin as it grows, which would be the basis to relate specific skin sections to time windows in the past.

Importantly, another potential source of cortisol is the epidermis itself. Several studies proved that mammalian skin express elements of the hypothalamo-pituitary–adrenal (HPA) axis such as the corticotrophin-releasing hormone (CRH), the CRH receptor-1 (CRH-R1), and key enzymes of corticosteroid synthesis [89,90]. This results in the local synthesis and release of cortisol as well as a negative feedback regulation on CRH expression. Therefore, skin can be considered functionally equivalent to the HPA axis [91]. The skin “HPA” may in fact coordinate the initial response to environmental stressors [92]. Therefore, cortisol concentrations measured here may not only reflect adrenal activity but cortisol locally derived [26,93].

Additionally, cortisol from the surface of the epidermis could be lost to sea water due to its polarity as suggested for baleen [27] and described in human hair immersion in water [94].

### 4.5. Implications of the Study and Future Research

To the best of our knowledge, this is the first evidence of cortisol measurements through a commercial enzyme immunoassay in the epidermis from any species of cetacean. Here, we propose a method to collect, process, and quantify epidermal cortisol concentrations in a standardized, simplified, and relatively economic way.

An assessment of cortisol in the epidermis may serve as a unique biomarker of the HPA activity over extended time periods in living cetaceans, avoiding the need of repeated samplings and the effect of non-recurrent short-term stress [26]. Moreover, the epidermis represents a more superficial tissue than blubber, and a sloughed portion can be collected non-invasively in some species and eventually away from the animal, thus benefiting the welfare impacts of research.

Cause–effect in relation to long-term stress in both captive and wild environments, as well as the significance of chronic vs. short-term exposure to stressors in these species, is an important study topic that could benefit from this technique. However, knowledge relevant to interpret cortisol levels in the epidermis of cetaceans is still missing, and to avoid misinterpretation, results should be approached cautiously.

Further studies should address some fundamental questions such as how GCs are incorporated into cetacean epidermis and how long they persist post deposition. To find this out, studies utilizing radioisotope-labelled GCs could be an excellent option [88,95]. Moreover, studies assessing the validity of epidermal cortisol concentrations as a welfare indicator for cetaceans are crucial and necessary. For instance, studies on epidermal cortisol concentrations in well-known captive individuals subjected to different stressors and changes in their welfare (e.g., changes in their social or physical environment, management, health, or behaviour) would generate valuable knowledge and allow studies of content, construct, and criterion validity. Stressor features and timing, and other potential stressors (not taken into account in the study design), should be recorded as accurately as possible [86]. Meanwhile, factors not directly related to physical or emotional stress like sex, age, or reproduction may alter baseline epidermal cortisol concentrations (as seen in other matrices) and should be characterized. Variations in hormone content in different sections of the epidermis and their capacity to represent cetaceans’ stress status could also be investigated (e.g., inner layer vs. outer layer, sloughed epidermis vs. cut epidermis, and full-depth epidermis vs. epidermis section).

Interestingly, other types of steroid hormones such as aldosterone, testosterone, or progesterone can also be found in cetacean epidermis [29]. The methanol-based extraction procedure described in this study can extract not only cortisol but other steroid hormones from epidermis matrices, as was conducted in other keratin matrices [85,96]. Thus, other commercial EIA assays could be validated as well for other steroid hormones such as progesterone, testosterone, and aldosterone, and potentially, a single epidermis sample and extraction procedure could allow the quantification of multiple steroid hormones in cetaceans.

## 5. Conclusions

We present a method for a feasible and non-invasive epidermis collection in captive common bottlenose dolphins and belugas. The study provides evidence that commercial EIA immunoassays can perform well in quantifying cortisol in cetacean epidermis. The individual had a significant effect on epidermal cortisol concentrations which may be related to individuals’ endocrinological status differences, while sex did not. We suggest that a sample mass of 20 mg has both good analytical results and a high probability of sampling in captive common bottlenose dolphins. Further studies are needed to evaluate the effect of body location on epidermis cortisol concentrations. However, we recommend standardizing the body location for epidermis sample collections within a study to reduce variability and avoid potential confusion regarding the data. Overall, our results suggest that epidermal hormone quantification potentially enables less or non-invasive and longer-term assessments of physiological stress response in cetaceans. To further develop epidermal cortisol concentrations as an indicator to be used in welfare and conservation research, it is necessary to perform physiological validations from individuals existing in a well-known welfare state.

## Figures and Tables

**Figure 1 animals-14-01377-f001:**
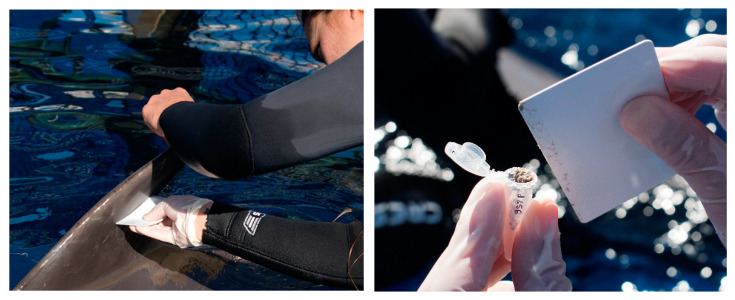
Collection of epidermis from the dorsal fin of a bottlenose dolphin in professional human care (*Tursiops truncatus*). The left image displays a trainer using a semi-rigid plastic card to scrape the fin after drying the area. The right image demonstrates how the epidermis sample is carefully placed into an Eppendorf tube to avoid contamination.

**Figure 2 animals-14-01377-f002:**
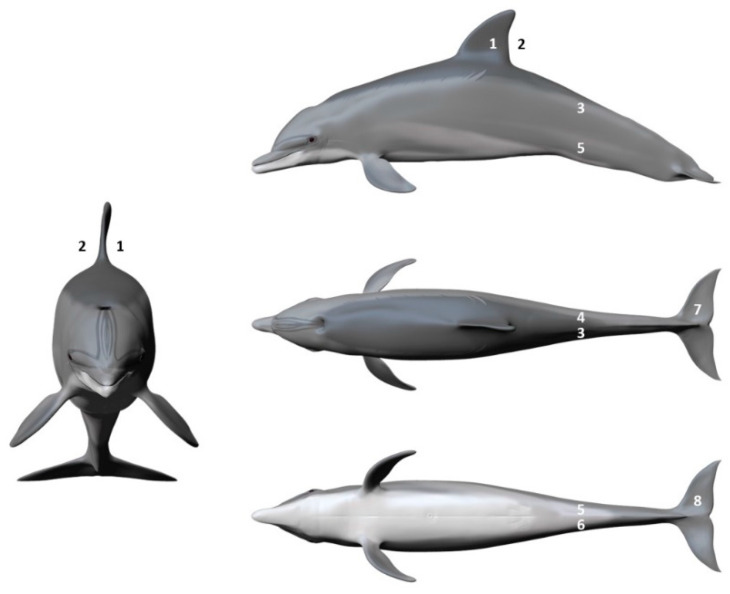
Body locations from which epidermis samples were obtained from ten common bottlenose dolphin individuals (*Tursiops truncatus*): (1) left dorsal fin; (2) right dorsal fin; (3) left dorsal peduncle; (4) right dorsal peduncle; (5) left ventral peduncle; (6) right ventral peduncle; (7) dorsal caudal fin; and (8) ventral caudal fin. Illustration credit: Emma Abad García, 2022.

**Figure 3 animals-14-01377-f003:**
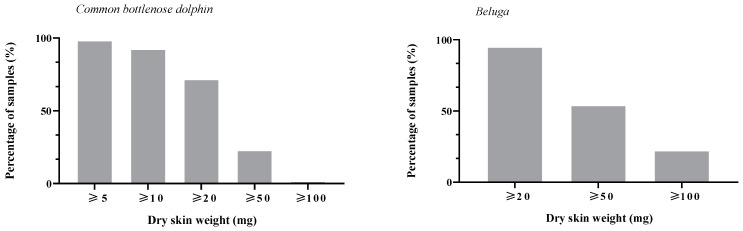
Percentage of collected epidermis samples equal or greater than 5, 10, 20, 50, and 100 mg of dry epidermis and 20, 50, and 100 mg of dry epidermis in common bottlenose dolphins (*Tursiops truncatus*) and belugas (*Delphinapterus leucas*), respectively.

**Figure 4 animals-14-01377-f004:**
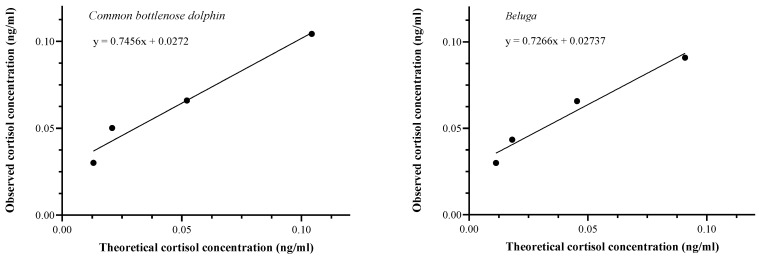
Biochemical validation of the enzyme immunoassay: correlation between observed and theoretical epidermal cortisol concentrations obtained in the dilution test. Left: common bottlenose dolphins (*Tursiops truncatus*; Pearson: r(3) = 0.96, *p* = 0.017); right: belugas (*Delphinapterus leucas;* Pearson: r(3) = 0.98, *p* = 0.017).

**Figure 5 animals-14-01377-f005:**
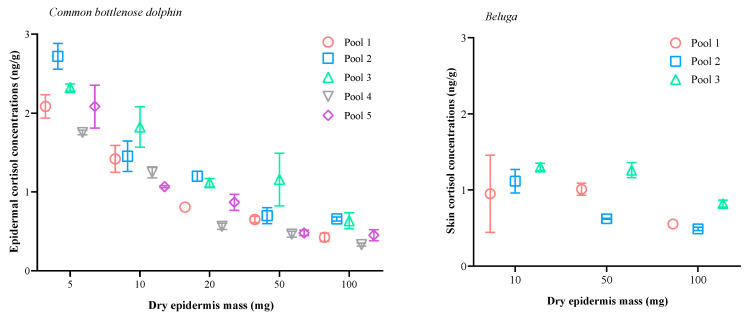
Relationship between sample mass (mean cortisol concentrations ± SEM) and epidermal cortisol concentration in duplicate subsamples of the pools obtained from pulverized epidermis from common bottlenose dolphins (*Tursiops truncatus*; **left**) and belugas (*Delphinapterus leucas*; **right**). Extractions were performed in duplicate for each sample mass.

**Figure 6 animals-14-01377-f006:**
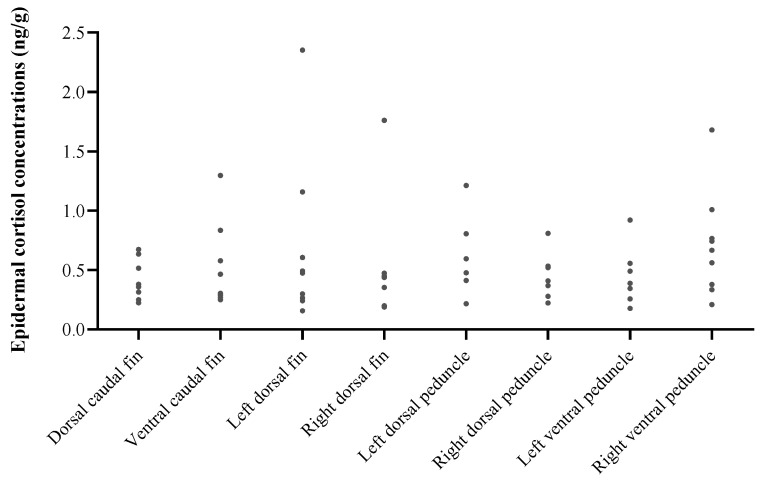
Epidermal cortisol concentrations across eight body sites in 10 common bottlenose dolphins (*Tursiops truncatus*): dorsal caudal fin, ventral caudal fin, left dorsal fin, right dorsal fin, left dorsal peduncle, right dorsal peduncle, left ventral peduncle, and right ventral peduncle.

**Figure 7 animals-14-01377-f007:**
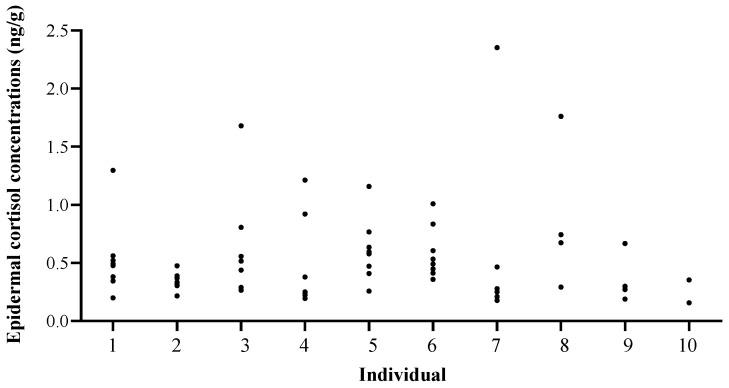
Epidermal cortisol concentrations across ten common bottlenose dolphins (*Tursiops truncatus*) in different body locations.

## Data Availability

The raw data supporting the conclusions of this article will be made available by the authors on request.

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
