# Peer review of "Validating a Non-Invasive Method for Assessing Cortisol Concentrations in Scraped Epidermal Skin from Common Bottlenose Dolphins and Belugas"

_animals, 2024, doi:10.3390/ani14091377_

Round 1

Reviewer 1 Report

Comments and Suggestions for Authors

Ln 3, 16, 17: common name for the species is Common Bottlenose Dolphin, and beluga without the “whale”. This also applies throughout the rest of the document. 

Ln 30: …concentrations in common bottlenose dolphin (Tursiops truncatus) and beluga (Delphinapterus

Ln 37: “Overall, we present a contribution…”

Ln 58, 74, 78: method of citation for Moberg and Mench, 2000?

Ln 92-94: difference between epidermal turnover and epidermal surface. Which is renewed every 73 days (epidermis and dermis?), and which is renewed every 2 hrs? clarify please.

Ln 105: Phocoena phocoena should be in italics, and citation method for Bechshoft

Ln 137. No need to mention scientific names again. 

Ln 141. Belugas included one male calf, one adult female and one adult male…

Ln. 145. …million liters of sea water.

Ln. 148, 153-154. Scientific names in italics.

Ln. 149. …Loligo spp.)

Ln. 152. …of sea water.

Ln 162. Between or during?

Ln. 167. I photo of the plastic card, or even the procedure would be good to add. 

Ln. 170. How was the skin in the plastic card moved into the narrow mouthof the sppendorf tube?

Figures look low resolution. Any chance to improve them?

Reviewer 2 Report

Comments and Suggestions for Authors

See uploaded document.

Comments on the Quality of English Language

Only minor revisions.

Reviewer 3 Report

Comments and Suggestions for Authors

This is basically the same paper as Bechsoft et al 2020. The only difference is the addition of the beluga whales and utilizing EIAs to measure cortisol. There is nothing unique about this publication and might be better suited as a methods paper.

Comments on the Quality of English Language

The English is fine.

Reviewer 4 Report

Comments and Suggestions for Authors

GENERAL COMMENTS:

The manuscript reads well and discusses an interesting topic on how to non-invasively determine cortisol levels in cetacean skin. The manuscript is well written and I only have very few comments/suggestions. The authors do clearly disclose that this method is not an optimal method to assess steroid hormone levels in captive and wild cetaceans. I appreciate that the authors clearly identify the issues with the methods but then discuss ways to improve the methods. Overall a great manuscript.

SIMPLE SUMMARY:

No comments.

ABSTRACT:

No comments

INTRODUCTION:

Lines 124-128: the authors should mention here that liquid chromatography with tandem mass spectrometry (LC-MS/MS)  is also commonly used and that more recent studies 2015+ have used this to quantify hormones in non-invasive samples from many different species.

MATERIALS AND METHODS:

Detailed description of methods and procedures. No comments.

RESULTS:

Line 322: there is a misspelling in this sentence.

Line 331: did not “find” replace “found”

DISCUSSION:

Line 385: replace with “did not vary” not “varied” 

Line 404: as mentioned above, you should also mention the LC-MS-MS method in the Introduction.

FIGURES AND FIGURE LEGENDS:

No comment. 

Comments on the Quality of English Language

Please see my comments as a few grammatical corrections are needed.
